# Electrochemical Performance of Orthorhombic CsPbI_3_ Perovskite in Li-Ion Batteries

**DOI:** 10.3390/ma14195718

**Published:** 2021-09-30

**Authors:** Nahid Kaisar, Tanmoy Paul, Po-Wei Chi, Yu-Hsun Su, Anupriya Singh, Chih-Wei Chu, Maw-Kuen Wu, Phillip M. Wu

**Affiliations:** 1Institute of Physics, Academia Sinica, Taipei 11529, Taiwan; nahid0kaisar@gmail.com (N.K.); paultanmoy00@gmail.com (T.P.); dr.pwchi@gmail.com (P.-W.C.); a810808a@gmail.com (Y.-H.S.); mkwu@phys.sinica.edu.tw (M.-K.W.); 2Research Center for Applied Science, Academia Sinica, Taipei 11529, Taiwan; anupriyas962@gmail.com (A.S.); gchu@gate.sinica.edu.tw (C.-W.C.); 3Department of Materials and Mineral Resources Engineering, National Taipei University of Technology, Taipei 10608, Taiwan

## Abstract

A facile solution process was employed to prepare CsPbI_3_ as an anode material for Li-ion batteries. Rietveld refinement of the X-ray data confirms the orthorhombic phase of CsPbI_3_ at room temperature. As obtained from bond valence calculations, strained bonds between Pb and I are identified within PbI_6_ octahedral units. Morphological study shows that the as-prepared δ-CsPbI_3_ forms a nanorod-like structure. The XPS analysis confirm the presence of Cs (3d, 4d), Pb (4d, 4f, 5d) and I (3p, 3d, 4d). The lithiation process involves both intercalation and conversion reactions, as confirmed by cyclic voltammetry (CV) and first-principles calculations. Impedance spectroscopy coupled with the distribution function of relaxation times identifies charge transfer processes due to Li metal foil and anode/electrolyte interfaces. An initial discharge capacity of 151 mAhg^−1^ is found to continuously increase to reach a maximum of ~275 mAhg^−1^ at 65 cycles, while it drops to ~240 mAhg^−1^ at 75 cycles and then slowly decreases to 235 mAhg^−1^ at 100 cycles. Considering the performance and structural integrity during electrochemical performance, δ-CsPbI_3_ is a promising material for future Li-ion battery (LIB) application.

## 1. Introduction

Lithium-ion battery (LIB) technology is considered as the most promising energy storage technology for a wide range of applications in portable electronic devices, electric vehicles (EV) and hybrid electric vehicles (HEV) [1]. In the United States, the Department of Energy has set a cost target for EV battery packs of 125 USD/kWh to be reached in 2022, which was in large part a driving mechanism for LIB technology [2]. Given the projected policy and economic environment, and the intrinsic energy density, long cycle life and stability during long cycling, LIB was made a contending choice as a commercialized energy storage system. Negatively charged lithium anodes play an important role in maintaining the high energy density of the battery [3,4]. In spite of having extremely high theoretical specific capacity (3860 mAhg^−1^) and low electrochemical potential (−3.04 vs. SHE), the safety risk of metallic Li hinders it from commercialization [5,6]. Commercial LIBs have graphite and lithium titanium oxide (LTO) as anode materials to achieve the market demand [7,8]. However, safety and energy density are a primary concern in those batteries. Graphite anodes face serious stability issues due to the heterogeneous dispersion of conductivity that results in the formation of imperfect solid electrolyte interphase (SEI) [9]. On the other hand, LTO exhibits outstanding stability in long-term cycling at very high C-rate while compromising safety of the battery [10]. However, low energy density and high operating voltage (1.54 vs. Li/Li^+^) made the commercialization of LTO impractical for a wide range of applications [11,12]. Recently, silicon (Si) has received a lot of attention as a superior anode material due to its high theoretical gravimetric capacity (4200 mAhg^−1^) and low operation voltage (0.4 V vs. Li/Li^+^) [13]. However, Si experiences 400% volume expansion during charge–discharge process, which affects the long cycle life due to increased probability for mechanical failure [14]. Indeed, given the environmental and material constraints, alternative anode materials for battery research should be developed to identify material choices that exhibit high energy density at low operating voltage while providing outstanding stability and safety. So far, various metal oxides, transition metal sulfides (MoS_2_, FeS_2_, Co_9_S_8_, VS_2_), carbon-based materials and Sn-based materials have been investigated as anode materials [15,16,17,18,19,20,21,22,23].

Organic and inorganic perovskites have a wide range of applications in the fields of solar cell, optoelectronics, laser and electrochromism [24,25,26,27,28,29,30]. Previously, scientists worked on the development of organic–inorganic perovskite materials as energy storage materials for Li-ion batteries [31,32,33,34,35,36,37]. However, these materials suffer from poor cycling performance and extremely low discharge capacity. The high mobility and Li^+^ storage capability of hybrid perovskites inspire us to use them as active materials. Previously, the orthorhombic CsPbI_3_ perovskite was successfully employed on top of Li metal to avoid the deadly dendrite, which causes failures to battery performance [38].

In this work, we report the δ-CsPbI_3_ perovskite as an active anode material for Li-ion batteries. We have prepared orthorhombic CsPbI_3_ by solution process at a constant temperature of 70 °C. The as-prepared anode material is incorporated in a coin cell against a Li metal. Further, we have investigated the Li storage capability and structural change of host material upon lithiation by the X-ray diffraction and ab initio density functional theory (DFT) calculation. The active material exhibits an initial discharge capacity of 151 mAhg^−1^ upon 40 mA g^−1^ current.

## 2. Experimental Section

### 2.1. Materials Preparation

The 0.1 M CsPbI_3_ solution was prepared by dissolving CsI_2_ and PbI_2_ (1:1) in a mixture of dimethylformamide (DMF) and dimethyl sulfoxide (DMSO) (vol 1:1) solvents. The as-prepared solution was stirred overnight at 70 °C. Finally, we obtained a yellowish powder which was further grinded using mortar and pestle to achieve a fine powder.

### 2.2. Material Characterization

The morphology of as-prepared active material was investigated using scanning electron microscopy (SEM, JEOL-Japan, JXA-840A). To observe the changes in electrodes after cycling, the coin cells were dissembled and the electrodes were washed using 1,3-dioxolane (DOL). Later, SEM was employed to observe the change in microscopic level. The crystallinity of δ-CsPbI_3_ was investigated using X-ray diffraction (XRD) technique using Philips X’Pert diffractometer equipped with a Cu Kα X-ray source (λ = 1.5406 Å) in the 2Ө range of 10°–50°. A JacobsV-670 UV–Vis spectrometer was employed to confirm the presence of absorption peaks of the δ-CsPbI_3_ and the band gap was calculated the Tauc plot. XPS (PHI 5000 Versa Probe and an Al Kα X-ray source = 1486.6 eV) was employed to study the binding energy of central metal ion Pb of δ-CsPbI_3_. Coin cells were assembled using δ-CsPbI_3_ electrode and after first discharge, the cell was opened and the electrode was washed with DOL and dried prior to transfer inside the XPS chamber.

### 2.3. Electrochemical Measurement

δ-CsPbI_3_ loaded electrodes were prepared by mixing 65% δ-CsPbI_3_ with 20% carbon black and 15% poly (vinylidene fluoride) (PVDF) binder. The powders were mixed in *N*-methyl-2-pyrrolidinone (NMP) solvent. As-obtained slurry was coated on Cu foil and dried overnight at 50 °C. Later, the electrode was cut into a 12 mm disk in order to prepare the coin cells. A 12 mm Li disk was used as the counter electrode. Electrode 1 M LiPF_6_ was prepared by mixing ethylene carbonate (EC) and dimethyl carbonate (DMC) (1:1 vol). A pristine Celgard separator was used in between two electrodes for safety reasons. CR2032-type coin cells were assembled in an Ar-filled glove box. Galvanostatic charge–discharge of the coin cells was performed using Think Power battery testing system. PARSTAT MC 1000 electrochemistry workstation was employed to measure the cyclic voltammetry (CV) and electrochemical impedance spectroscopy (EIS). CV tests were performed on the cells in a voltage window of 0.1–3 V and at a scan rate of 0.1 mV s^−1^. EIS of the cells were performed with an AC amplitude of 10 mV and the frequency range was from 0.01 Hz to 1 MHz.

### 2.4. Theoretical Treatment

The distribution function of relaxation times (DFRTs) was calculated by the Impedance Spectroscopy Genetic Programming (ISGP) program using the impedance spectra after different cycles [39,40,41,42,43,44]. As the DFRT approach is applicable only in the Kramers–Krönig (KK) relations compatible regime, the supporting figure (Appendix A, Appendix A) suggests that we can safely identify three peaks within the KK compatible regime (around 0.1 Hz). A similar approach is also adopted elsewhere [39,40,41,45]. The Rietveld refinement of the XRD pattern was performed by FullProf software package, the Fourier maps were calculated using GFourier and the crystal structure was plotted using VESTA [46,47].

The electronic structure calculation was performed using the Quantum ESPRESSO (QE) distribution, based on density-functional theory, periodic boundary conditions, plane-wave basis sets and pseudopotentials [48,49]. The GBRV ultrasoft pseudopotentials at a plane-wave cutoff of 40 Ry and a charge-density cutoff of 200 Ry were selected for all the calculations with a Gamma-centered k-point mesh [50]. The structure optimization was carried out using a conjugate-gradient algorithm until the forces on each atom were less than 0.01 eV Å^−1^ and the total energy was converged up to 10^−5^ eV. To study the lithium intercalated structures of 0.5, 1, 2, 3 and 4, supercells of LiCs_8_Pb_8_I_24_, LiCs_4_Pb_4_I_12_, Li_2_Cs_4_Pb_4_I_12_, Li_4_Cs_4_Pb_4_I_12_ and Li_8_Cs_4_Pb_4_O_12_ structures with their stable configuration have been considered, respectively.

## 3. Results and Discussion

1 M δ-CsPbI_3_ was prepared by mixing CsI_2_ and PbI_2_ in DMF and DMSO solvent which results in a yellow solution (Appendix Aa). The mixture was stirred overnight inside a N_2_ filled glove box at 70 °C. We received a yellowish CsPbI_3_ powder as our final active material (Appendix Ab). Figure 1 exhibits the crystal structure of δ-CsPbI_3_ and diffraction peaks assigned to orthorhombic structure or δ-phase as determined by the Rietveld refinement [51]. Although most of the perovskites are air-sensitive, δ-CsPbI_3_ is extremely stable in air [52]. δ-CsPbI_3_ maintains its crystallinity as confirmed by the present solution method. According to previous reports, CsPbI_3_ exists in four crystal phases: α, β, γ and δ [53]. Out of all the studied phases, the δ phase is stable in air, and an orthorhombic structure gives sufficient space for Li intercalation–deintercalation [52]. The diffraction pattern of the δ-CsPbI_3_ at room temperature has been indexed with orthorhombic phase (space group Pnma) having lattice parameters *a* = 10.4595(3) Å, *b* = 4.8034(15) Å and *c* = 17.7754(4) Å, α = β = γ = 90° with a cell volume of 893.055 Å^3^ which is in accordance with ICSD File #250744. The convergence of the refinement is achieved with the residuals of R_B_ = 7.87%, R_F_ = 5.75% and χ^2^ = 7.34, and the refined structural parameters are shown in Table 1. The XRD patterns at different lithiation potentials, determined by CV measurement shown in Appendix A, have a smaller number of peaks in comparison with the pristine powder. Preferred orientation of the Bragg peaks in 2θ around 43° and 50° (Appendix A), and Appendix A (red curve), supports the conversion reactions. For instance, the peaks at 42.93°, 50.07°, 27.23° in panel (b), and 42.95°, 50.10° in panel (c) can be attributed to δ-CsPbI_3_; 25.88°, 31.06° are associated with PbI_2_ and 36.02°, 52.02° in panel (d) are from PbLi, as confirmed from ICSD Files #250744, #104762, #68819, #24265, respectively. This suggests the coexistence of both pristine and conversion reacted phases at different potentials. The Rietveld refinements of the XRD patterns for both pristine and after 100 cycles (Appendix A with orthorhombic phase) suggest an increase in unit cell volume (896.747–893.055 = 3.692 Å^3^) indicating lithiation in the cell as well as shallow charging/discharging processes. Furthermore, the lithiation potential calculation by DFT supports this conjecture (see later section). However, the pristine structure changes due to prolonged cycling as well as reaction times of lithiation (Appendix A and Appendix A). Additionally, the presence of Li_x_PF_y_ moieties as confirmed by the XPS studies further justifies the lithiation process (see Appendix A and Appendix A, and Equation (1)).

The perovskite structure of δ-CsPbI_3_ has distorted PbI_6_ octahedral with a distortion index of 0.025 Å and two Pb-I1, three Pb-I3 and single Pb-I2 distances of 3.232(7) Å, 3.303(7) Å, 3.409(7) Å and 3.060(6) Å, respectively (Appendix A). The average I-I distances are calculated as 4.188(9) Å to 4.803(15) Å. In this context, the electron density distributions along (110) plane are calculated to find any discrepancy between the observed and calculated structure factors. As observed from Appendix A, the positive electron densities at two highly localized I1 sites justify the proper fitting of the model. Nevertheless, the contour lines suggest the ionic bonding between I sites. The bond-valence parameters have been calculated using the Zachariasen formula to be 1.923 v.u., 1.029 v.u., 1.081 v.u., 0.928 v.u. and 1.106 v.u. for Pb, I1, I2, I3 and Cs sites, respectively. Interestingly, the variation in the bond-valence parameter of I changes due to different atomic coordinates and occupancies (see Table 1). These deviations from their original valence states indicate the strained bonding inside the polyhedral units, and these have a direct effect on battery performance as observed here and elsewhere [54].

Anisotropic thermal factors for the same elements are shown below (Table 2).

Figure 2a,b shows regular and uniform nanorod-like structures with an average length of 5–7 μm and diameter of ~ 400 nm of δ-CsPbI_3_. The average particle size is 5.5 μm which is calculated from particle size distribution histogram (Appendix A). An absorption peak is observed between 425–450 nm when UV-vis spectra was collected for δ-CsPbI_3_ (Appendix A). Tauc plot shows a band gap of 2.644 eV as obtained from the optical absorption spectrum. DFT-based density of states (DOS) is used to investigate how the lithium ion affects the electronic structure of the δ-CsPbI_3_ perovskite. From Appendix A, δ-CsPbI_3_ shows a band gap of 2.132 eV similar to that obtained for CsPbBr_3_ [55]. Our results suggest that Perdew–Burke–Ernzerhof (PBE) underestimates the experimental band gap up to 21%. The top-view SEM image (Figure 2c) shows that the CsPbI_3_ is uniformly distributed. Figure 2d shows ~15 μm electrode coated on the Cu foil. An XPS survey spectrum is performed to confirm the compositions of CsPbI_3_, Cs, Pb and I_2_ at their corresponding binding energies (Figure 2e).

The impedance measurements were carried out in the pristine cell as well as after several charge–discharge cycles (Figure 3a). We have also plotted the impedance data for a Li‖Li symmetric cell for comparison. It is observed that the overall impedance response contains a single semicircular arc with spike-like extensions at low frequencies. With increasing charge–discharge cycles, the diameter of the semicircular arc decreases and shows an abrupt change after maximum 60 cycling. To understand the electrochemical phenomenon qualitatively, the impedance data have been modeled by ISGP, neglecting the capacitive diffusive regime at low frequencies. The program computes the distribution function of relaxation times which comprises several peaks justifying some electrochemical phenomena. As observed from Figure 3b, the Distribution Function Relaxation Times (DFRTs) for all the conditions show three peaks within the experimental frequency range. A tiny peak at the highest frequencies corresponds to the Ohmic drop due to electrolytic resistances (Peak P1). The peak within the 10^2^ to 10^4^ Hz regime corresponds to lithium metal, which is related to the charge transfer processes along the metal surface (Peak P2) [56,57]. The peak shifts towards low frequencies with initial cycling but vanishes after 100 cycles, indicating loss of the reactivity of the lithium metal. Additionally, the charge transfer resistance decreases with increasing cycles, justifying good electronic transport (Appendix A). At pristine condition, due to the initial SEI layer at anode, lithium activity maintains an equilibrium which dilutes with increasing cycle numbers and as a result, the resistance decreases. Secondly, the peak within 10–10^−2^ Hz corresponds to the charge transfer of lithium ions at the anode/electrolyte interface and exhibits strong frequency dependence (Peak P3) [56]. This charge transfer process becomes slower with increasing cycle numbers. A close inspection shows that the peak P3 for cycles 2 and 3, unlike cycles 1 and 70 (Figure 3a inset), has broader time distribution which is due to small variation in their impedance plots. It seems that the charge transfer processes are important in explaining the state-of-health of the batteries and can be used as representative fingerprinting for such high-capacity behavior.

The XPS spectra in Figure 4a reveal the electrochemical interaction of lithium with δ-CsPbI_3_. Both spectra contain peaks from Cs (3d, 4d), Pb (4d, 4f, 5d) and I (3p, 3d, 4d) and with a decrease in their intensities (except O and F) in the material after charging–discharging cycle, suggesting a coating of surface film on the active electrode [58,59]. A tiny peak at 532 eV of O 1s is observed which could be due to Li alkoxy species: CH_3_OLi (from reduction of DMC) and (CH_2_OLi)_2_ (from reduction of EC) compounds as obtained (Figure 4a) [60]. Secondly, the peak becomes sharper after the discharging condition. A peak at 286 eV corresponds to unavoidable C 1s peak (C-C). XPS spectra exhibit the Pb^2+^ states and the binding energies 138.8 and 143.7 eV, corresponding to 4f_7/2_ and 4f_5/2_, respectively (Figure 4b). Previously, it was reported that due to the influence of external ions/factors, there might be change in oxidation state of Pb^2+^ or redistribution of electron charges around Pb^2+^ [55,61]. The binding energies shifted from 138.8 to 137.7 eV and 143.7 to 142.77 eV, respectively. The absence of I 4d_5/2_ peak at 49 eV and I 3d_5/2_ and 3d_3/2_ around 630 eV confirm that the discharging effect is associated with iodine. Overall, as obtained from the XRD refinement results, the PbI_6_ octahedra are associated with the lithiation process. As expected, no LiF peaks are detected in the pristine electrode. Another important feature is that the F 1s spectra at 688 eV was observed after discharged (Appendix A). The F 1s spectra consists of two peaks around 685 eV (due to LiF) and 687 eV due to Li_x_PF_y_ with the following reduction reactions [60]:(1)$$LiPF_{6}+\left(5+x-y\right)e^{-}+\left(5+x-y\right)Li^{+}\to \left(6-y\right)LiF+Li_{x}PF_{y\;}$$

The Li 1s peak at 56 eV can be observed (LiF) (Appendix A) and the P 2p peak (138 eV) is masked with Pb 4f, inferring Li_x_PF_y_ moieties. On the other hand, a relatively higher O 1s peak suggests Li_x_PF_y_O_z_ moieties. Since the electrode is washed thoroughly with DMC before the measurements, the peaks relating to F 1s and P 2p may not be due to original LiPF_6_. Thus, the surface of the electrode is modified by different species such as LiF, Li_x_PF_y_ and Li_x_PF_y_O_z_ due to the first discharged cycle.

Figure 5a represents the cyclic voltammetry (CV) of Li-ion coin cells with δ-CsPbI_3_ as anode materials. The CV is performed within a voltage range of 0.1–3 V at a scan rate of 0.1 mV s^−1^. During delithiation, two peaks at 0.58 V and 0. 7 V vs. Li/Li^+^ are noted, which have been reported for MAPbBr_3_ [62]. Furthermore, during lithiation, the peaks between 0.4–0.6 V can be attributed to Li_x_Pb such as LiPb and Li_2_._6_Pb as identified previously for PbO_2_, PbO and Pb electrodes [63]. It is noted that similar characteristic features of alloying and dealloying have been reported for 2D and 3D hybrid perovskites [64]. Structural defragments due to alloying are also noted from SEM images (Appendix A). Concerning the peak positions and their variations in CV and XPS spectra, the following electrochemical reaction is proposed: CsPbI_3_ + Li^+^ + e^−^ →Li_x_CsPbI_3_ (during discharging). Theoretically, to determine the lithiation potential in CsPbI_3_, Li-ion is considered to be inserted in the primitive orthorhombic unit cell of CsPbI_3_ (20 atoms) with its different stoichiometries. Since in the pristine structure, Cs, Pb and I-ions occupy 4c sites, we have chosen 8d, 4c, 4b and 4a as interstitial lithiation sites. The geometry optimization calculations suggest that the ground state configuration energy is the same for all the sites with the same Li atoms per unit cell (as listed in Appendix A). Based on the CV curve (Figure 5a), the peaks between 0.4 V–0.6 V can be associated with Li_x_Pb such as LiPb, Li_2_._6_Pb as previously observed for Pb-, PbO_2_- and PbO-based electrodes, whereas the peak around 1.4 V is due to Li_x_CsPbI_3_ (Figure 6a). To identify the quantity of the intercalated lithium ion per unit cell, we have performed geometry optimization in QE with x = 0.5, 1, 2, 3 and 4 Li-ions only at 8d site of the host structure. Figure 6a shows a difference in the insertion voltages at each x referring to a single-phase and one step mechanism of lithiation, and a variation in the lattice parameters is observed (Figure 6b). For instance, the lithiation reaction having 0.5 Li-ion will be CsPbI_3_ + 0.5Li^+^ + e^−^ → Li_0_._5_CsPbI_3_. Nevertheless, the simulated XRD patterns of Li_0_._5_CsPbI_3,_ LiCsPbI_3,_ Li_2_CsPbI_3,_ Li_3_CsPbI_3_ and Li_4_CsPbI_3_ are similar to that of the pristine structure (orthorhombic phase). Indeed, the parent structure is an indirect band gap semiconductor, but with an insertion of 0.5 Li per unit cell, the structure becomes semi-metallic (Appendix A). However, the band gap widens with an increase of lithium content (Appendix A). The formation energy and the average Li intercalation potential curves in Figure 6 show that with increasing lithiation, the magnitude of the formation energy increases and the average Li intercalation potential decreases. As the formation energy of lithiation is less than 0 eV, the intercalation is definitely possible. However, the discharge curve in CV suggests that conversion reaction relating Li_x_Pb starts around 0.6 V (Figure 5a). Thus, we can confirm that both intercalation and conversion have occurred during the whole discharge process, and due to the conversion process, a structural phase transition may exist (Appendix A and Appendix A). Comparing Figure 5a,b and Figure 6, the kinks in the discharge CV curve around 1.4 V and 1.0 V can be attributed to Li_0.5_CsPbI_3_ and Li_4_CsPbI_3_, respectively. It is noted that similar insertion and conversion reactions during discharging are reported for the W_18_O_49_ anode [65,66]. Nevertheless, the unit cell volume expands for the lithiated structures (Figure 6b). It is observed that a small volume change is a direct consequence of the low-density perovskite structure of allotrope CsPbI_3_, facilitating Li-ion movement with minimal lattice distortion. The CV results indicate that the electrochemical reduction of δ-CsPbI_3_ is reversible. Additionally, the peaks at charging condition do not show any voltage drift, signifying a steady formation of SEI layer as well as no electrolyte oxidation during the cell operation [67]. Figure 5b exhibits dQ/dV profiles ranging from 0.01 to 3 V without any loss of capacitance during the first cycle. This feature establishes its applicability as an anode material in comparison with graphite [68]. The post-cycling cell (after 100 charge–discharge cycles) was tested in similar condition, in order to observe the impact of lithiation/delithiation on the CV of the active material. Appendix A shows the reversible CV profile of δ-CsPbI_3_ after 100 cycles inferring negligible loss of active material during long cycling, and as a result, the capacity is maintained around the same as that after 75 cycles. The SEM image of δ-CsPbI_3_ electrode after 100 cycles observed shows a smooth and integrated surface (Appendix A). Appendix A is the cross-sectional image of the electrode after 100 charge–discharge cycles showing the coating remains intact with the Cu foil. This study suggests that the δ-CsPbI_3_ electrode has good integrity throughout a long cycle life and that there is no loss of active materials during cycling. Later, we disassembled the cell and washed and dried the electrode to observe the XRD peak. Appendix A shows major peaks for δ-CsPbI_3_ maintained after 100 cycles.

Figure 7 exhibits the long-term cycling performance of Li-ion battery using δ-CsPbI_3_ as active material at 40 mA g^−1^. The δ-CsPbI_3_ electrode suffers from a capacity drop in the first six charge–discharge cycles from 151 mAhg^−1^ to 149.8 mAhg^−1^. From seventh cycle, the capacity started to increase steadily with an increase rate of 2.12 mAhg^−1^ per cycle to reach a maximum of ~275 mAhg^−1^ at 65 cycles. After that, it drops to ~240 mAhg^−1^ at 75 cycles and then slowly decreases to 235 mAhg^−1^ at 100 cycles. This phenomenon suggests that Li-ion accessibility increases inside the active material along with oxygen containing functional groups with increasing the cycling number. This can be observed in Appendix A with reversible CV of active material after 70 cycles. Additionally, this could be due to a large surface area (SEM) and high oxygen adsorption at the surface as supported by the XPS spectrum (Figure 4a). There are a few valid reasons behind a slight increase in discharge capacity which are as follows: (a) due to increase in activation of the active materials, (b) activation of defects in the active materials, and (c) enlargement of active materials’ interatomic space after a few charge–discharge cycles (as understood from increment from lattice parameters) [69,70,71]. Another possibility is that there is a conversion reaction happening at low voltages which could suppress the intercalation, and due to the conversion reaction, the structural defragment is obtained (Appendix A). Overall, the capacity has increased Li-ion accessibility inside the active material along with oxygen containing functional groups with an increase in the cycling number. The Coulombic efficiency was maintained throughout the cycling performance, which suggests a reversibility of Li insertion/disinsertion during electrochemical process. This cycling performance reveals that Li^+^ can intercalate–deintercalate during the charge–discharge process, and it is a reversible process [72,73]. This suggests the Li^+^-ion movement during cycling was smooth and avoided thicker SEI formation [74]. Furthermore, as suggested by the dQ/dV plot (Figure 5b), there is a negligible SEI layer in comparison with that of graphite [74].

We have shown the cycling performance at different C-rates to evaluate the cycling stability of the active material. As displayed in Appendix A, the cell exhibits an initial discharge capacity of 181 mAhg^−1^ when discharged at 0.1 C-rate. After tested through harsh charge–discharge conditions at different C-rates, the cell retains ~84% of the initial discharge capacity, which shows outstanding cycling performance of the CsPbI_3_ as an active anode material.

## 4. Conclusions

We have successfully introduced δ-CsPbI_3_ as an active anode material for Li-ion battery. The perovskite structure of δ-CsPbI_3_ has distorted PbI_6_ octahedra as calculated by the X-ray refinement. Ionic bonding between I sites are observed along with a partial vacancy up to 0.03%. We propose that the presence of the functional species has a beneficial effect in terms of discharge capacity. Specifically, the charge transfer processes along the Li metal surface and that at the anode/electrolyte interface are identified, and both of them show constant frequency shifts with increasing cycling. We conclude that both intercalation and conversion have occurred during the whole discharge process using ex situ XRD, CV and first-principles calculations. The electrochemical performance of δ-CsPbI_3_ shows outstanding potential as a promising anode material for commercialized Li-ion batteries. We need more extensive studies to stabilize performance while at the same time increasing their capacity.

## Figures and Tables

**Figure 1 materials-14-05718-f001:**
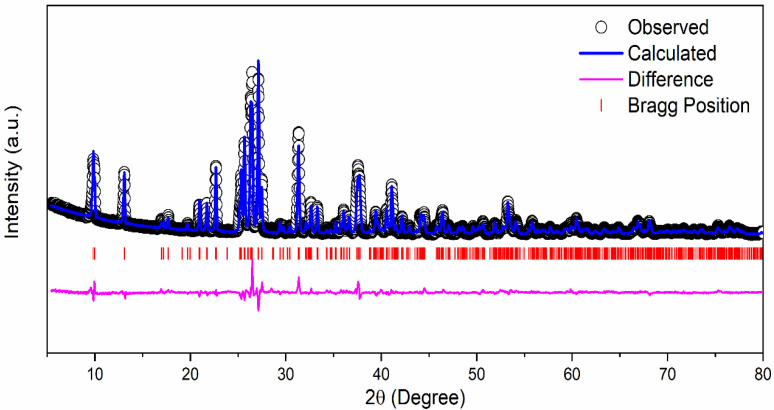
Rietveld refinement of the XRD pattern of δ-CsPbI_3_ at room temperature.

**Figure 2 materials-14-05718-f002:**
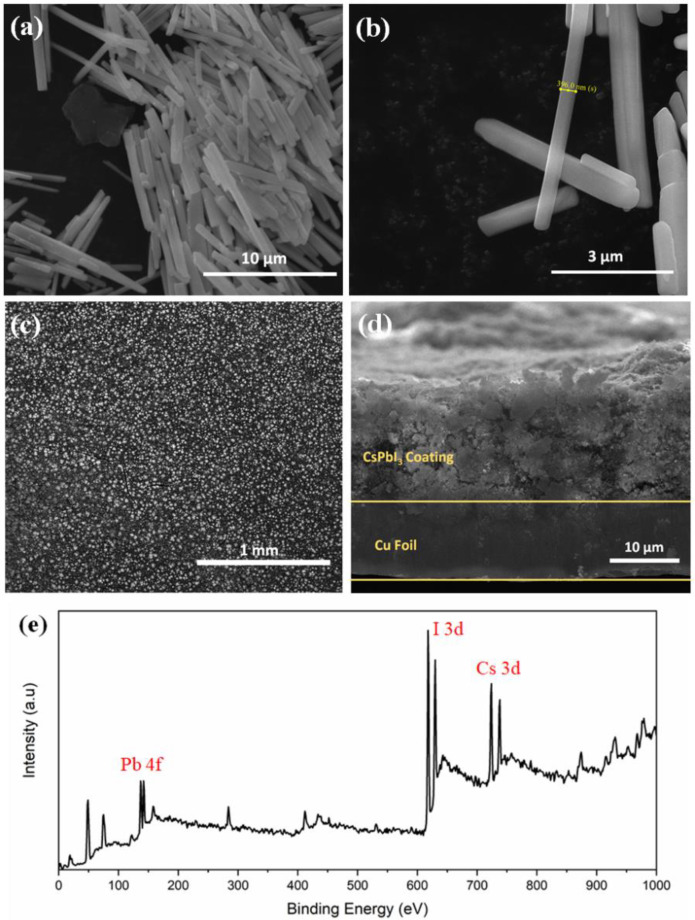
Morphology analysis. (**a**,**b**) FESEM image of as-prepared δ-CsPbI_3_ using solution process at different magnifications. (**c**) Top view FESEM image of electrode using δ-CsPbI_3_ and PVDF, carbon black pasted on Cu foil. (**d**) Cross-sectional FESEM image of as prepared electrode on Cu foil. (**e**) XPS survey spectrum of δ-CsPbI_3_ as prepared by solution process.

**Figure 3 materials-14-05718-f003:**
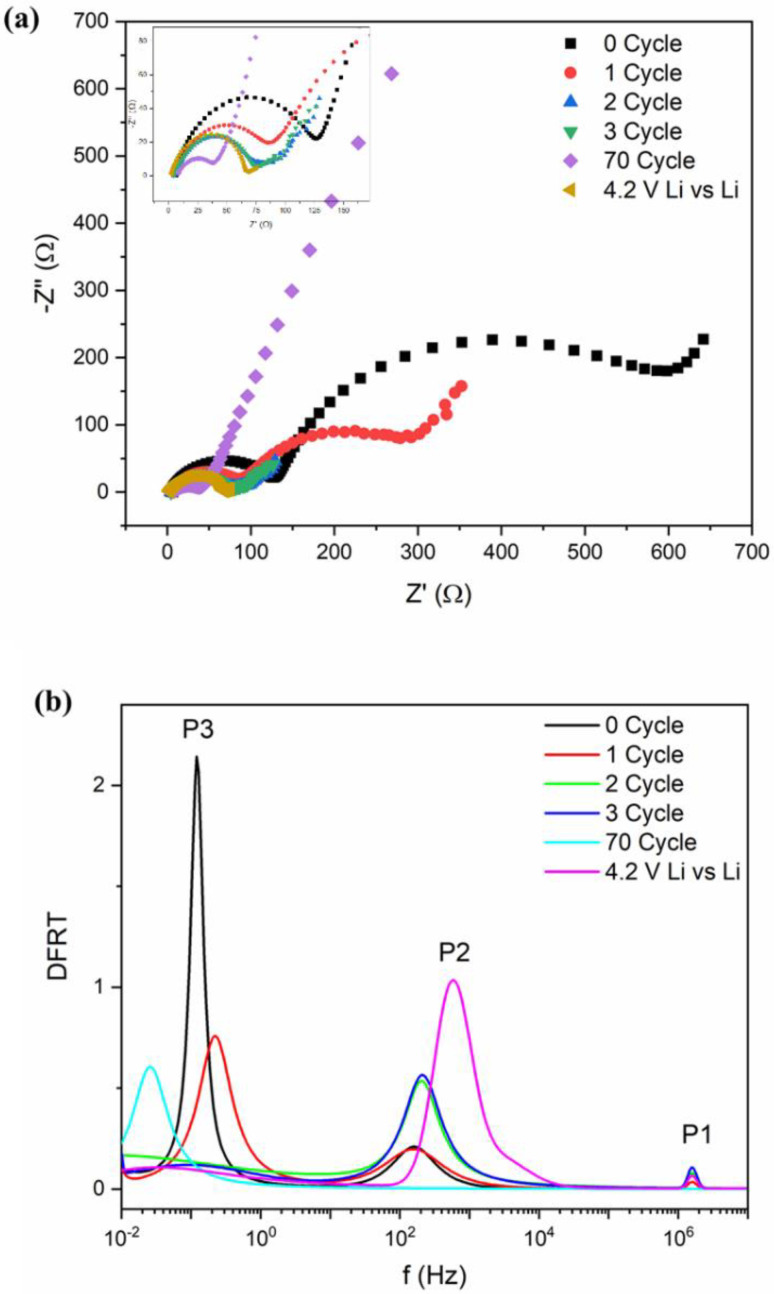
(**a**) Nyquist plots of impedance after different cycles along with symmetric Li cell. (**b**) DFRTs for Li-ion batteries after different cycling measurements. Inset of (**a**): zoom-in figure of impedance.

**Figure 4 materials-14-05718-f004:**
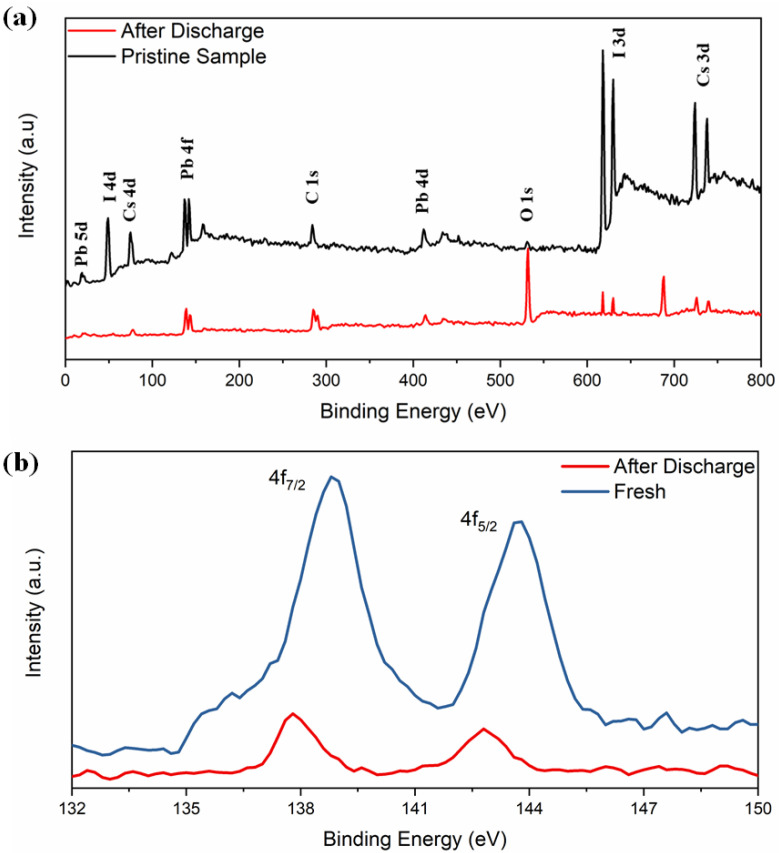
(**a**) Full scan XPS spectra of CsPbI_3_, (**b**) XPS spectra of Pb 4f before and after electrochemical discharging process.

**Figure 5 materials-14-05718-f005:**
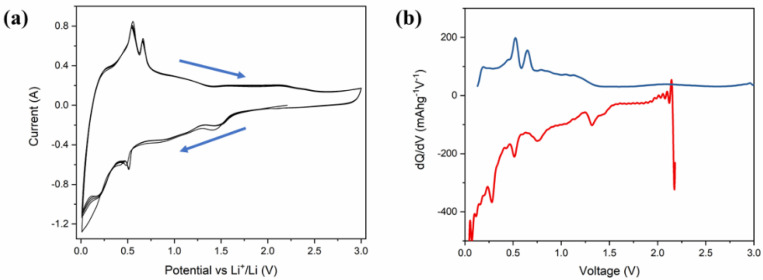
Electrochemical performance of δ-CsPbI_3_ active materials in Li-ion half-cell. (**a**) Cyclic voltammetry trace of a δ-CsPbI_3_ half-cell within a voltage window of 0.1–3.0 V, recorded at a scan rate of 0.1 mV s^–1^. (**b**) Incremental capacity dQ/dV vs. cell potential as recorded at a rate of 40 mA g^−1^ for first electrochemical cycle.

**Figure 6 materials-14-05718-f006:**
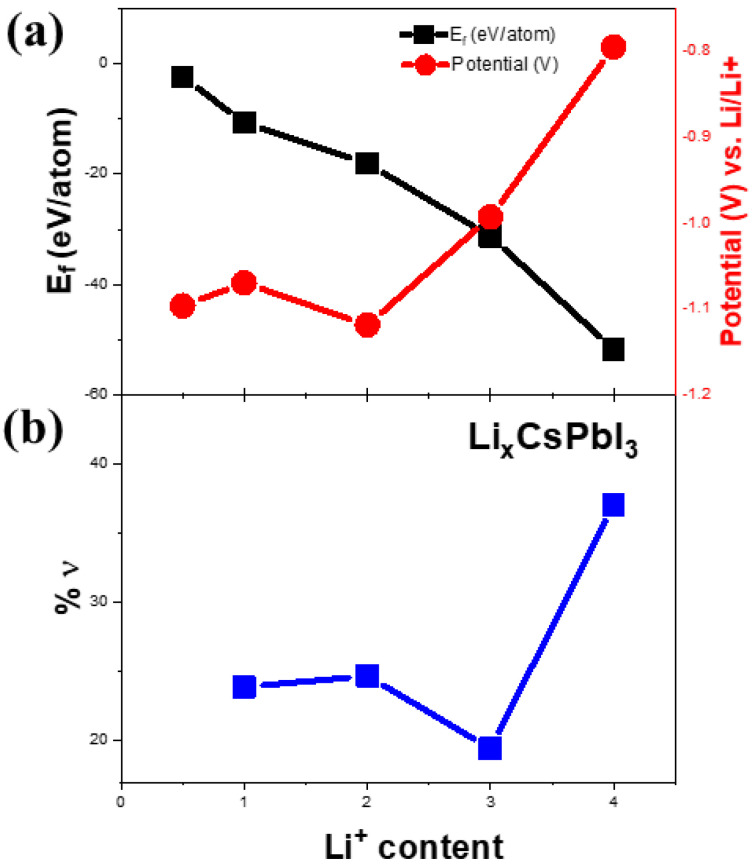
(**a**) Calculation of formation energies as a function of lithium content. The right vertical axis represents the lithiation potential as a function of lithium content in (**a**). Theoretically calculated volume change (%ν) as a function of lithium content is shown in (**b**).

**Figure 7 materials-14-05718-f007:**
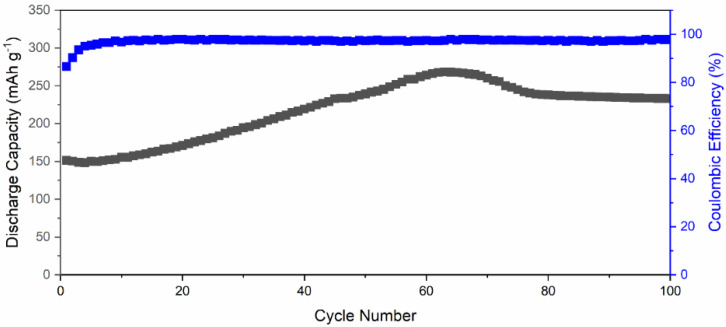
Long-term cycling performance of a Li-ion battery featuring δ-CsPbI_3_ as active materials, tested at 0.1C-rate.

**Table 1 materials-14-05718-t001:** At room temperature.

Label of Atom	Atom	Wyckoff Site	Atomic Coordinates			Occupancy
			x	y	z	
Pb	Pb	4c	0.16099	0.2500	0.43864	1
I1	I	4c	0.16396	0.2500	0.00343	0.995
I2	I	4c	0.29759	0.2500	0.28639	0.974
I3	I	4c	0.03533	0.2500	0.61563	1
Cs	Cs	4c	0.41541	0.2500	0.66999	1

**Table 2 materials-14-05718-t002:** Anisotropic thermal factors for the same elements.

Label of Atom	B_11_	B_22_	B_33_	B_12_	B_13_	B_23_
Pb	0.00148	0.04421	0.00015	0	0.00088	0
I1	0.00302	0.01320	0.00154	0	0.00173	0
I2	−0.00049	0.05040	0.00014	0	0.00023	0
I3	0.00127	0.04586	0.00086	0	−0.00028	0
Cs	−0.00077	0.07601	0.00194	0	0.00076	0

## Data Availability

The data presented in this study are available on request from the corresponding author.

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
