# Peer review of "Electrochemical Performance of Orthorhombic CsPbI3 Perovskite in Li-Ion Batteries"

_materials, 2021, doi:10.3390/ma14195718_

Round 1

Reviewer 1 Report

Major revisions

In the proposed manuscript entitled “Electrochemical Performance of Orthorhombic CsPbI3 Perovskite in Li-ion batteries”, N. Kaisar and co-authors reported an electrochemical and structural study of a CsPbI3 perovskite as active anode material in Lithium-ion batteries.

The pristine system and the charge/discharge system have been characterized by morphological and structural studies, as well as XPS and electrochemical measurements.

The XRD studies and CV measurements, together with the theoretical calculations are in my opinion a strong basis to support the intercalation and conversion of lithium ions in the perovskite structure upon several charge/discharge process.

The  electrochemical and structural analysis provides an adequate support for the use of CsPbI3 material as an efficient anode in Li batteries.

However, in my opinion, the chemical analysis based on X-ray photoemission spectroscopy is not conclusive in giving a portrait on the chemical changes occurring during the charge/discharge cycles. A deeper study of Pb and Li oxidation states should be performed, in order to understand the chemical evolution of the system. My suggestion is to perform photoemission experiments on samples with different stages of degradation. This can be done,  for instance, taking advantage of the high flux, tunability and resolution of synchrotron facilities.

Moreover, the text should be reorganized, separating structural, morphological, and  electrochemical studies, in order to have more linear description of the entire set of results.

The overall English should be also improved.

For these reasons, I recommend the publication of the submitted paper only after major revisions on the organization of the text, and the photoemission analysis.

Minor Revisions

Page 5, line 9: change the phrase in “…as observed in figure S11 with the increasing of cycles number…”

The image in figure 4 is reversed, change it.

Author Response

Response to Reviewer# 1

Q1. In the proposed manuscript entitled “Electrochemical Performance of Orthorhombic CsPbI3 Perovskite in Li-ion batteries”, N. Kaisar and co-authors reported an electrochemical and structural study of a CsPbI3 perovskite as active anode material in Lithium-ion batteries.

The pristine system and the charge/discharge system have been characterized by morphological and structural studies, as well as XPS and electrochemical measurements.

The XRD studies and CV measurements, together with the theoretical calculations are in my opinion a strong basis to support the intercalation and conversion of lithium ions in the perovskite structure upon several charge/discharge process.

The electrochemical and structural analysis provides an adequate support for the use of CsPbI3 material as an efficient anode in Li batteries.

However, in my opinion, the chemical analysis based on X-ray photoemission spectroscopy is not conclusive in giving a portrait on the chemical changes occurring during the charge/discharge cycles. A deeper study of Pb and Li oxidation states should be performed, in order to understand the chemical evolution of the system. My suggestion is to perform photoemission experiments on samples with different stages of degradation. This can be done, for instance, taking advantage of the high flux, tunability and resolution of synchrotron facilities.

Answer:  Thanks to the reviewer for valuable comments and suggestions on our manuscript. The texts, and figures has been separated to make a linear description of the results. All the changes have been highlighted with yellow colour. Later suggestions by the reviewer is quite challenging to follow within the timeframe allowed by the editor. Our Synchrotron facility is not available due to maintenance and ongoing long holidays in Taiwan.

Q2. Moreover, the text should be reorganized, separating structural, morphological, and electrochemical studies, in order to have more linear description of the entire set of results.

The overall English should be also improved.

Answer: The experimental section has been rearranged in the revised version

We have tried to improve the English in the manuscript.

Q3. Page 5, line 9: change the phrase in “…as observed in figure S11 with the increasing of cycles number…”

Answer: The required changes have been made in the revised version.

Q4. The image in figure 4 is reversed, change it.

Answer: The required changes have been made in the revised version.

Reviewer 2 Report

The authors prepared orthorhombic CsPbI3 and they used it as an anode material for Li-ion battery. They investigated the Li storage capability and structural change of host material.

In my opinion the topic of the manuscript is interesting and it is worth of publication. Before publishing major changes are required. My specific comments:

Major revisions:

1) The manuscript should be prepared using the Microsoft Word template or LaTeX template (available on the journal's website).

2) The structure of the manuscript is incorrect. The Experimental section (which is after the Conclusions chapter in the reviewed manuscript) should be placed after Introduction as the "Methods and materials" chapter.

3) The manuscript should contain the required statements.

4) The bibliographic references are not in the style of the Materials. Furthermore, several articles are grouped in one number (e.g. [3] - two articles, [11] - nine articles).
They should be separated - each article should have its own individual number. 

5) The conclusions should be redrafted. Now it is rather a summary of the activities and not conclusions. Please go straight to the point.

Minor revisions:

6) As the readers of the publication will be specialists from various fields of science, not all acronyms and designations of chemical compounds will be known to them. Therefore, they should be clarified the first time they are used in the text. This applies to e.g. CV, SEI, DMF, DMSO.

7) On pages 2 and 3 it is written: "shown in Table I ...; see Table I ...". There is no table with this number in the manuscript and in the supplementary information. 

8) Table 3: The title of the table is incomplete. Moreover, the numbering of tables should start with the number 1 (not 3).

9) There is an error in the Figure 4 - it shows the mirror image of the spectra.

Author Response

Response to Reviewer# 2

Q1.  The manuscript should be prepared using the Microsoft Word template or LaTeX template (available on the journal's website).

Answer: The required template has been used in the revised version.

Q2.  The structure of the manuscript is incorrect. The Experimental section (which is after the Conclusions chapter in the reviewed manuscript) should be placed after Introduction as the "Methods and materials" chapter.

Answer: Thanks to the reviewer for valuable comments and suggestions on our manuscript. We have followed as the reviewer suggested and made changes in the manuscript. We moved the “experimental section” after the “introduction” and write whole manuscript in the word format which is available on journal’s website. We changed the reference format and rewrite the conclusion part.

Q3.  The manuscript should contain the required statements.

The bibliographic references are not in the style of the Materials. Furthermore, several articles are grouped in one number (e.g. [3] - two articles, [11] - nine articles).
They should be separated - each article should have its own individual number. 

The conclusions should be redrafted. Now it is rather a summary of the activities and not conclusions. Please go straight to the point.

Answer: The required changes have been amended.

The conclusion has been rewritten as “We have successfully introduced δ-CsPbI3 as an active anode material for Li-ion battery. The perovskite structure of δ-CsPbI3 has distorted PbI6 octahedra as calculated by the X-ray refinement. Ionic bonding between I sites is observed along with a partial vacancy up to 0.03%. We propose that the presence of the functional species has a beneficial effect in terms of discharge capacity. Specifically, charge transfer processes along the Li metal surface and that at the anode/electrolyte interface are identified, and both of them show constant frequency shift with increasing cycling. We concluded that both intercalation and conversion have occurred during the whole discharge process using ex-situ XRD, CV, and first-principles calculations. The electrochemical performance of δ-CsPbI3 shows outstanding potential as a promising anode material for commercializing the Li-ion batteries. We need more extensive studies to stabilize the performance, at the same time increase the capacity.”

Q4. As the readers of the publication will be specialists from various fields of science, not all acronyms and designations of chemical compounds will be known to them. Therefore, they should be clarified the first time they are used in the text. This applies to e.g. CV, SEI, DMF, DMSO.

Answer: The required changes have been amended.

Q5. On pages 2 and 3 it is written: "shown in Table I ...; see Table I ...". There is no table with this number in the manuscript and in the supplementary information. 

Answer: The required changes have been amended.

Q6.  Table 3: The title of the table is incomplete. Moreover, the numbering of tables should start with the number 1 (not 3).

Answer: The required changes have been amended.

Q7. There is an error in Figure 4 - it shows the mirror image of the spectra.

Answer: Figure 4 has been replaced with the right one.

Round 2

Reviewer 1 Report

I thanks the authors for their revised manuscript. They have positevely responded to my suggestions and the reviosions made improved the quality of the manuscript.

For this reasons I recommend the paper for puiblication.

Reviewer 2 Report

In the resubmitted work the authors have addressed my comments and recommendations.

I believe the manuscript has been sufficiently corrected and improved. In my opinion it is suitable for publication.